# Metabolic Profiles of Whole Serum and Serum-Derived Exosomes Are Different in Head and Neck Cancer Patients Treated by Radiotherapy

**DOI:** 10.3390/jpm10040229

**Published:** 2020-11-13

**Authors:** Anna Wojakowska, Aneta Zebrowska, Agata Skowronek, Tomasz Rutkowski, Krzysztof Polanski, Piotr Widlak, Lukasz Marczak, Monika Pietrowska

**Affiliations:** 1Institute of Bioorganic Chemistry Polish Academy of Sciences, 61-704 Poznan, Poland; astasz@ibch.poznan.pl; 2Maria Sklodowska-Curie National Research Institute of Oncology, Gliwice Branch, 44-101 Gliwice, Poland; aneta7zebrowska@gmail.com (A.Z.); agata.w2012@gmail.com (A.S.); Tomasz.Rutkowski@io.gliwice.pl (T.R.); Piotr.Widlak@io.gliwice.pl (P.W.); 3Wellcome Sanger Institute, Wellcome Genome Campus, Hinxton, Cambridgeshire CB10 1SA, UK; kp9@sanger.ac.uk

**Keywords:** head and neck cancer, exosomes, serum, radiotherapy, metabolomics, GC/MS

## Abstract

Background: In general, the serum metabolome reflects the patient’s body response to both disease state and implemented treatment. Though serum-derived exosomes are an emerging type of liquid biopsy, the metabolite content of these vesicles remains under researched. The aim of this pilot study was to compare the metabolite profiles of the whole serum and serum-derived exosomes in the context of differences between cancer patients and healthy controls as well as patients’ response to radiotherapy (RT). Methods: Serum samples were collected from 10 healthy volunteers and 10 patients with head and neck cancer before and after RT. Metabolites extracted from serum and exosomes were analyzed by the gas chromatography–mass spectrometry (GC–MS). Results: An untargeted GC–MS-based approach identified 182 and 46 metabolites in serum and exosomes, respectively. Metabolites that differentiated cancer and control samples, either serum or exosomes, were associated with energy metabolism. Serum metabolites affected by RT were associated with the metabolism of amino acids, sugars, lipids, and nucleotides. Conclusions: cancer-related features of energy metabolism could be detected in both types of specimens. On the other hand, in contrast to RT-induced changes observed in serum metabolome, this pilot study did not reveal a specific radiation-related pattern of exosome metabolites.

## 1. Introduction

Head and neck cancer (HNC) is the sixth most common malignancy worldwide. The incidence of HNC exceeds half a million annually and accounts for approximately 6% of all cancer cases worldwide [1,2]. Although over the last decade we have observed an improvement in the treatment of HNC, there is still a need for new biomarkers of this type of cancer because, since the tumor location and classical staging remain the major criteria of the treatment selection, and molecular heterogeneity of HNC [3]. Radiotherapy (RT), used either alone or in combination with other treatment modalities (surgery, chemotherapy, or immunotherapy) is the major modality in the HNC treatment. The major benefit of RT is a well established local control of the tumor. However, ionizing radiation induces damage to the adjacent healthy tissues, which is reflected at the systemic level in body fluids [4,5,6,7]. Hence, detection in the patient’s blood of a molecular fingerprint of the body’s response to the treatment is another important aspect of HNC diagnostics, which could potentially enable the monitoring and prediction of radiation toxicity.

Various types of “omics” studies (genomics, transcriptomics, proteomics, metabolomics) using different sources of samples (blood, urine, saliva, tissues) uncovered molecules and genes of potential use as clinical biomarkers [8,9]. Cancer cells’ metabolism differs from one of the healthy cells and it is considered as the closest footprint of a cancer phenotype. This is why metabolomics studies are among the fastest-growing areas of cancer research in recent decades. Recent studies revealed disparities in the metabolite profile between diseased and normal states as well as between miscellaneous types of cancer or various stages of the disease. Therefore, altered metabolic pathways in various cancer systems might be used to identify biomarkers in terms of diagnosis, prognosis, or treatment schedule choice [10,11]. The NMR-based metabolomics study revealed an increased level of glucose, ketone bodies, ornithine, asparagine, and 2-hydroxybutyrate while decreased levels of citric acid cycle (TCA cycle) intermediates (citrate, succinate, and formate), lactate, alanine, and other gluconeogenic amino acids in the sera of patients with HNC [12]. Another GC/MS-based metabolomic analysis of serum and tissues of HNC patients revealed different metabolite profiles in patients with different treatment outcomes. In patients with disease relapse, the serum levels of metabolites related to the glycolytic pathway (especially glucose, ribose, fructose) were higher while serum levels of amino acids (lysine and trans-4-hydroxy-L-proline) were lower than in samples of patients without disease relapse [13]. Hence, one could conclude that the altered energy metabolism, mostly the switch from the TCA cycle to aerobic glycolysis known as the Warburg effect, is characteristic for patients with HNC [12,13,14,15]. Moreover, a few studies addressed therapy-induced changes in metabolic profiles of HNC, revealing compounds whose levels were associated with the treatment escalation (e.g., of the radiation dose during radiotherapy) or the intensity of treatment toxicity [16,17,18]. However, the knowledge about molecular mechanisms involved in radiation-induced changes of the patient’s metabolome remains limited.

In the present study, the GC/MS approach was applied to profile the serum metabolites of HNC patients who underwent RT to uncover the metabolome changes induced by radiation. Furthermore, we included in the study another emerging biospecimen—exosomes circulating in patients’ blood. Exosomes are virus-sized (50–150 nm) vesicles of endosomal origin released by the majority of cell types, either normal and cancerous [19]. Exosomes and other classes of extracellular vesicles (EVs) play an essential role in cancer biology, being the key mediators of communication between cells [20,21]. EVs present in the blood and other biofluids represent an interesting type of so-called liquid biopsy, which is an emerging source of potential biomarkers with applicability in treatment personalization [22,23]. There is a growing evidence for the increased level of EVs in the biofluids of cancer patients as well as the radiation-induced enhancement of exosome secretion [24,25]. Even though literature data support the important role of transcriptome and proteome content of cancer-related EVs, much less is known about their metabolome component [20]. Similarly, the data regarding radiation-induced changes in the EVs’ cargo refer mainly to its transcriptome and proteome [26,27]. Hence, through searching for cancer-related and RT-induced changes in the serum metabolome of HNC patients, we aimed to address the metabolite profiles of serum-derived EVs.

## 2. Materials and Methods 

### 2.1. Samples Collection 

Ten patients with squamous cell carcinoma located in pharynx regions (6 males and 4 females, aged between 49 and 71 years) treated by the continuous accelerated irradiation (CAIR) scheme with a daily fraction dose of 1.8 Gy to the total dose of 64–72 Gy were included in the study. Blood samples were collected before RT (cancer pre-treatment sample A) and one month after the end of RT (cancer post-treatment sample B). The control group constituted of ten age- and sex-matched healthy volunteers (control sample C). This study was approved by the appropriate local Ethics Committee (NRIO; approval no. 1/2016) and all participants provided informed consent indicating their conscious and voluntary participation. 5 milliliters of blood was collected into an anticoagulant-free tube (Becton Dickinson, Franklin Lakes, NJ, USA; 367955), incubated for 30 min at 20 °C then centrifuged at 1000× *g* for 10 min at 4 °C. The serum (supernatant) was transferred to clean tubes stored at −80 °C until analysis.

### 2.2. Exosomes Isolation and Characterization

The method for the isolation of exosomes from small amounts of serum was established and optimized in our laboratory as described previously [28]. Briefly, exosomes were isolated from serum (500 µL) by differential centrifugation (1000× *g* and 10,000× *g* for 10 and 30 min, respectively, at 4 °C) and filtration through a 0.22 µm filter followed by the size exclusion chromatography (SEC). SEC was performed using hand-packed columns (BioRad) filled with 10 mL of Sepharose CL-2B (GE Healthcare), conditioned previously with phosphate buffer saline (PBS). Consecutive fractions (500 µL each) were collected and characterized for EV enrichment (fraction #8 was used for further analyses). The size of vesicles in the SEC fractions was evaluated by the dynamic light scattering (DLS) using Zetasizer Nano-ZS90 instrument (Malvern Instruments, Malvern, UK) and by transmission electron microscopy. Exosome markers CD63 and CD81 were analyzed by Western blots as reported in detail elsewhere [28]. The concentration of proteins in the analyzed samples was assessed using the PierceTM BCA Protein Assay kit (Thermo Fisher Scientific, Waltham, MA, USA; 23225) according to the manufacturer’s instructions. 

### 2.3. Metabolite Extraction

Two-hundred microliters of 80% MeOH was added to 25 μL of serum sample. In the case of exosomes, 2 mL of 100% MeOH was added to 500 µL of the selected SEC fraction. The mixture was vortexed and centrifuged for 5 min followed by sonication for 10 min. The mixture was placed at −20 °C for 20 min and after that centrifuged for 10 min at 23,000× *g* at 4 °C. The supernatant was transferred to a new tube and evaporated in a SpeedVac concentrator (CentriVap Concentrator, Labconco, USA). The dried extract was then derivatized with 25 μL of methoxyamine hydrochloride in pyridine (20 mg/mL) at 37 °C for 90 min with agitation. The second step of derivatization was performed by adding 40 μL of MSTFA (*N*-Trimethylsilyl-*N*-methyl trifluoroacetamide) and incubation at 37 °C for 30 min with agitation. Samples were subjected to GC/MS analysis directly after derivatization.

### 2.4. GC–MS Analysis

The GS/MS analysis was performed using TRACE 1310 gas chromatograph connected with TSQ8000 triple-quad mass spectrometer (Thermo Scientific, Waltham, Massachusetts, USA). A DB-5MS bonded-phase fused-silica capillary column (30 m length, 0.25 mm inner diameter, 0.25 μm film thickness) (J&W Scientific Co., Folsom, California, USA) was used for separation. The GC oven temperature gradient was as follows: 70 °C for 2 min, followed by 10 °C/min up to 300 °C (10 min), 2 min at 70 °C, raised by 8 °C/min to 300 °C and held for 16 min at 300 °C. For sample injection, a PTV (Programmable Temperature Vaporization) injector was used in a range of 60–250 °C, transfer line temperature was set to 250 °C, and source to 250 °C. Spectra were recorded in m/z range of 50–850 in EI+ mode with an electron energy of 70 eV. Raw MS-data were converted to abf format and analyzed using MSDial software package v. 3.96. To eliminate the retention time (Rt) shift and to determine the retention indexes (RI) for each compound, the alkane series mixture (C-10 to C-36) was injected into the GC/MS system. Identified artifacts (alkanes, column bleed, plasticizers, MSTFA, and reagents) were excluded from further analyses. Obtained normalized (using total ion current (TIC) approach) results were then exported to Excel for pre-formatting and then used for statistical analyses.

### 2.5. Statistical and Chemometric Analyses

Differences between independent samples were assessed using the T-test, Welch test, or U-Mann–Whitney test, dependent on the normality and homoscedasticity of data (assessed via the Shapiro–Wilk test and Levene test, respectively). For paired samples, the paired *t*-test or Wilcoxon test were used based on the normality of the difference distribution. In each case, the Benjamini–Hochberg protocol was used for the false discovery rate (FDR) correction. However, due to the small sample size, none of the differences remained significant after the FDR correction. Hence, the effect size analysis was employed to overcome this problem [29]. For independent samples, the rank-biserial coefficient of correlation (RBCC; an effect size equivalent of the U-Mann–Whitney test) was applied; the effect sizes ≥ 0.3 and ≥ 0.5 were considered medium and high, respectively [30]. For paired samples, the paired *t*-test derived Cohen’s d effect size was applied; the effect sizes ≥ 0.5 and ≥ 0.8 were considered medium and high, respectively [31]. Principal component analysis (PCA) and hierarchical cluster analysis (HCA) based on the Euclidean distance method were performed to illustrate general similarities between samples. Metabolic pathways were associated with differentiating compounds that showed high and medium effect sizes using the quantitative enrichment analysis on the MetaboAnalyst platform (https://www.metaboanalyst.ca/MetaboAnalyst/ModuleView.xhtml). Obtained enriched pathways and their connections together with statistical information were further analyzed in Cytoscape. The DyNet addon was used to compare two networks and find interacting nodes [32]; the fold enrichment and significance of enrichment (FDR) were coded by the size and color of nodes, respectively.

## 3. Results

Extracellular vesicles (EVs) isolated from serum by size exclusion chromatography were characterized by their size and the presence of specific biomarkers. The SEC fraction #8 was enriched in vesicles, in which size was estimated in a range between 50 and 150 nm by the DLS measurement (with the maximum at 100–120 nm) (Figure 1A). The size of the isolated vesicles was confirmed by transmission electron microscopy (TEM) (Figure 1B). Furthermore, the presence of exosome biomarkers, tetraspanins CD63, and CD81, was confirmed in the same fraction by Western blot analysis (the same proteins remained undetected in the whole serum) (Figure 1C). Considering their specific size and the presence of exosome-specific biomarkers, vesicles present in the analyzed fraction were called exosomes for simplicity, yet other subpopulations of the small EVs could be present in this fraction.

The GC–MS-based approach was used to profile the metabolites in the whole serum and the corresponding serum-derived exosomes of HNC patients, in either pre-treatment (A) and post-treatment (B) samples, or samples of matched healthy controls (C). In general, the untargeted approach allowed to identify 182 metabolites in serum samples and 46 metabolites in exosome samples, of which 33 metabolites overlapped; the complete list of 195 identified compounds is presented in Appendix A. Figure 2 illustrates the distribution of different classes of small metabolites identified by GC–MS in serum and serum-derived exosome samples. Among the most abundant classes of metabolites common for serum and exosomes were fatty acids, sugar alcohols, and carboxylic acids (22%, 15%, and 12% of all identified compounds, respectively). It is noteworthy that amino acids that were the largest group of metabolites in serum samples that were markedly less frequent in exosome samples (21% vs. 7% of all identified compounds, respectively, which corresponded to 40 and 3 compounds). All identified metabolites were used to perform the unsupervised clustering of samples. The metabolite composition of the whole serum enabled the relatively good separation of all three groups of samples using either the principal component analysis (Figure 3A) or the hierarchical cluster analysis (Figure 4A). Interestingly, control samples C were more similar to cancer pre-treatment samples A than to cancer post-treatment samples B, which indicated the additional putative treatment-related differential component. In contrast, neither the PCA or the HCA type of analysis allowed the separation of corresponding groups when samples of serum-derived exosomes were analyzed (Figure 3B and Figure 4B).

In the next step, we detected specific metabolites for which abundances were significantly different between groups. First, we looked for compounds that differentiated cancer patients (pre-treatment samples A) from healthy individuals (control samples C). There were 27 compounds for which serum levels were markedly different (large effect size; RBCC effect size ≥ 0.5) between control and cancer samples. These included 12 upregulated metabolites (four amino acids, four fatty acids, two purines, one glycerolipid, and lactose) and 15 downregulated metabolites (three carboxylic acids, three purines, three sugars, two fatty acids, serotonin, acetyl-hexosamine, isoleucine, and phosphate) in cancer samples, listed in Table 1. Furthermore, there were 18 cancer-upregulated and 38 cancer-downregulated compounds where differences showed a medium effect size (RBCC effect size ≥ 0.3) (Appendix A). On the other hand, there were only a few compounds whose abundance was significantly different in serum-derived exosomes from healthy controls and cancer patients. 1-Hexadecanol was markedly upregulated while citric acid, 4-hydroxybenzoic acid, and propylene glycol were markedly downregulated (large effect size) in exosomes from cancer patients (Table 1). Moreover, there were seven metabolites where differences showed no medium effect size, including myo-inositol, linoleic acid, succinic acid, and glyceric acid downregulated in cancer samples (Appendix A). Metabolites for which levels were different between control and cancer samples (either a large effect size or medium effect size) were annotated with their corresponding metabolic pathways. Interestingly, the overrepresented pathways associated with metabolites discriminating cancer patients and healthy controls (i.e., cancer-specific pattern) in both whole serum and serum-derived exosome samples included ones involved in energy production (citric acid cycle, Warburg effect, pyruvate metabolism, mitochondrial electron transport chain) and inositol metabolism. Pathways associated specifically with serum metabolites included the metabolism of amino acids, sugars, and lipids. On the other hand, pathways associated with metabolites specific for serum-derived exosomes included the oxidation of fatty acids and ketone body metabolism (Figure 5A).

Then, we looked for metabolites with an abundance that was different in serum and serum-derived exosomes of cancer patients between pre-RT samples A and post-RT samples B, to allow the detection of changes related to radiotherapy. There were 12 compounds with serum levels that were markedly different (large effect size; Cohen’s d effect size ≥ 0.8) between pre-RT and post-RT cancer samples. These included four metabolites that were upregulated (including hypotaurine and serotonin) and eight metabolites that were downregulated in post-RT serum samples, listed in Table 2. Furthermore, there were 29 RT-upregulated and 12 RT-downregulated compounds where differences showed a medium effect size (Cohen’s d effect size ≥ 0.5) (Appendix A). In marked contrast, only two metabolites detected in serum-derived exosomes (glycerol and cholesterol) showed reduced levels (medium effect size) in post-RT samples. Finally, metabolites with an abundance different in the pre-RT and post-RT samples (either large effect size or medium effect size) were annotated with their corresponding metabolic pathways. Over represented pathways associated with metabolites with serum level affected by RT included those involved in the metabolism of different classes of compounds (amino acids, sugars, nucleotides, lipids, and biogenic amines), which indicated multifaceted effects of radiation on the serum metabolome profile (Figure 5B).

## 4. Discussion

Serum-derived exosomes, an emerging type of liquid biopsy, are a potential source of biomarkers. However, their metabolite compartment is less characterized compared to proteome or miRNome [20]. Here, we compared the metabolite profiles of whole human serum and serum-derived exosomes, and found significantly fewer compounds in the latter specimen. This difference could be caused by both a lower number of compounds present in vesicles per se (i.e., putatively lower molecular complexity) or their lower concentration, which hindered their detection by the method used in our approach. Hence, a direct comparison of metabolic pathways associated with compounds present in the whole serum and serum-derived vesicles could be compromised by this discrepancy. However, it has to be emphasized that the major metabolic hallmark of cancer—the modified energy metabolism could be detected in both specimens. In head and neck cancer, as in many other types of cancers, tumor cells can alter their energy metabolism by switching from the citric acid cycle (TCA cycle) to the aerobic glycolysis and oxidation of fatty acids as a backup mechanism for energy production [16], a phenomenon which is known as the Warburg effect [33]. Our study confirmed that metabolites associated with processes involved in energy metabolism, including glycolysis, gluconeogenesis, Warburg effect, TCA cycle, pyruvate metabolism, and mitochondrial electron transport chain showed different levels in samples of HNC patients and healthy controls. Importantly, features associated with this characteristic cancer phenotype were observed in both whole serum and serum-derived exosomes. Noteworthy, however, different types of cells and tissues, both cancerous and normal, release exosomes circulating in the blood and regulate the metabolome of the whole serum. Nevertheless, pathways associated with metabolites specific for serum-derived exosomes of cancer patients included oxidation of fatty acids and ketone body metabolism. The beta-oxidation of fatty acids and increased lipolysis, which is reflected as the accumulation of ketone bodies, was reported in HNC patients as a potential backup mechanism for energy production [12]. Previous studies reported that molecules involved in fatty acids transport and storage as well as lipolysis and fatty acids oxidation are enriched in EVs and suggested that fatty acid transport from cell to cell and across cell membranes could be mediated by EVs [34,35]. Hence, a specific role of serum EVs in the transmission of mediators associated with cancer-related lipid metabolism deserves further attention.

Our study revealed that RT affected the serum levels of several amino acids, biogenic amines, sugars, nucleotides, lipids, and fatty acids, which mirrored potential RT-induced changes in a plethora of metabolic pathways ongoing in a patients’ body. It is noteworthy that different radiation-related mechanisms might contribute to metabolic changes observed in samples collected one month after the end of RT, including toxicity induced by radiation in normal tissues and a reduced number of cancer cells. It was previously reported that the altered metabolism of amino acid plays an important role in the response of HNC patients to RT [36]. For example, Boguszewicz and co-workers [4] demonstrated that a decreased serum level of alanine, the main substrate for gluconeogenesis during fasting and cachexia, correlated with the acute radiation toxicity-associated weight loss in HNC patients undergoing RT. The whole-body response to irradiation frequently involves molecules associated with oxidative stress and inflammation [18]. Hence, it is noteworthy that hypotaurine, which is involved in protection against oxidative stress as an effect of RT [37], was significantly elevated in post-RT serum samples. Moreover, RT-induced changes in the serum level of phospholipids potentially associated with the inflammatory response and disruption of plasma membranes were previously reported in samples of HNC patients [38,39]. Here, we found that compounds associated with lipid metabolism (e.g., phosphatidylethanolamine and phosphatidylcholine biosynthesis) were affected in post-RT serum samples, which confirmed the general RT-related metabolic phenotype. Interestingly, very few RT-related changes were detected in the metabolic profile of serum-derived exosomes. This was in contrast to the significant radiation-induced changes observed at the level of the proteome [40] and miRNome [41] of exosomes released by HNC cells. Exosomes released by irradiated cells are known mediators of radiation bystander effect and other aspects of radiation-related cell-to-cell signaling [42]. Hence, the potential role of metabolites in exosomes-mediated radiation-related signaling should be addressed in further studies.

## 5. Conclusions

In this pilot study, we compared the metabolite profiles of the whole serum and serum-derived exosomes in healthy controls and patients treated with RT due to a head and neck cancer aiming to reveal cancer-related features (by the comparison of cancer and control samples) and RT-related features (by the comparison of cancer pre-RT and post-RT samples). We found that the metabolite profile of serum-derived exosomes is putatively less complex and consists of fewer components than that of the complete serum. However, cancer-related features of energy metabolism were detected in both types of specimens, which confirmed the feasibility of cancer biomarkers based on exosome metabolites. On the other hand, in contrast to RT-induced changes observed in serum metabolome, this pilot study did not reveal a specific pattern of radiotherapy-related changes in exosome metabolites. Hence, further metabolomics study with a larger cohort of individuals treated with RT is necessary to validate a hypothetical radiation signature of serum exosomes.

## Figures and Tables

**Figure 1 jpm-10-00229-f001:**
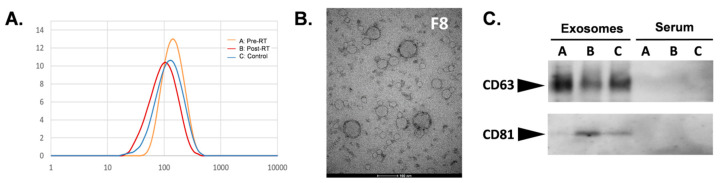
Characterization of serum-derived exosomes. Analysis of the size of vesicles in the size exclusion chromatography (SEC) fraction #8 by the dynamic light scattering (**A**) and transmission electron microscopy (**B**). (**C**) Western blot analysis of CD63 and CD81 in whole serum and serum-derived exosomes (fraction #8) for the three groups of samples (**A**: pre-radiotherapy (RT), **B**: post-RT, **C**: control).

**Figure 2 jpm-10-00229-f002:**
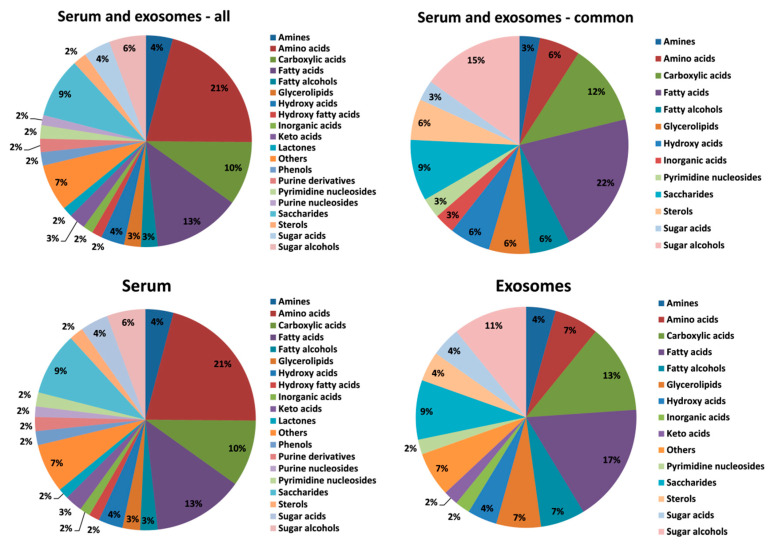
The relative contribution of different classes of metabolites present in serum and serum-derived exosomes (metabolites detected in all types of analyzed samples were considered).

**Figure 3 jpm-10-00229-f003:**
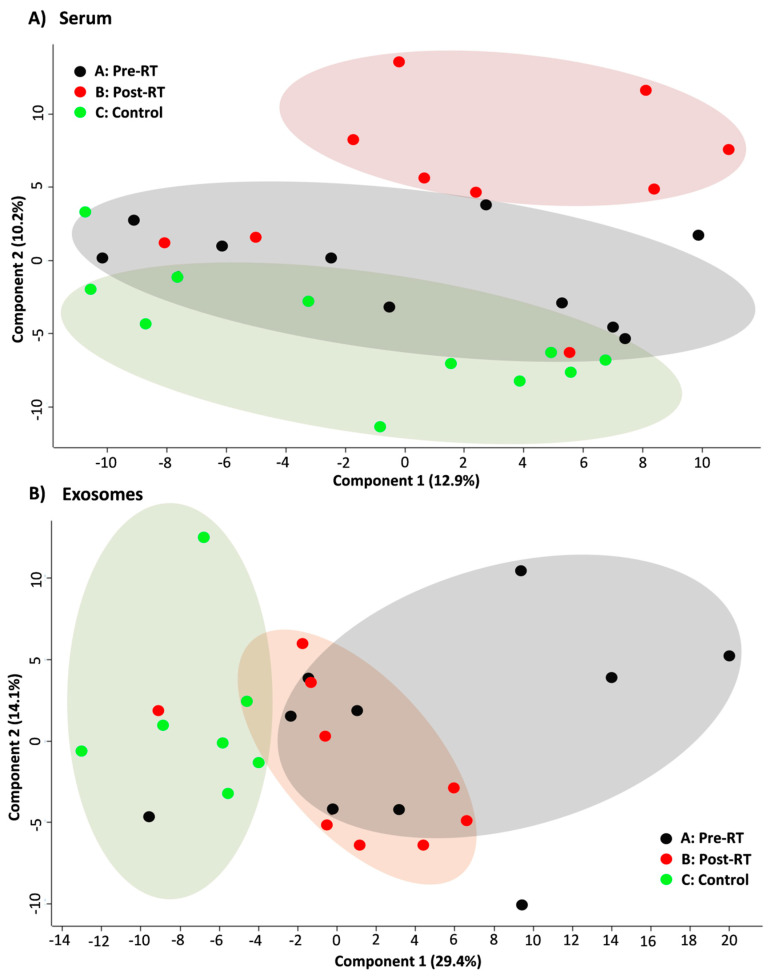
PCA score plots showing the clustering of cancer pre-RT samples A, cancer post-RT samples B, and control samples C. Shown are two the first components responsible for 23.1% of the variability of the serum samples (panel **A**) and 43.5% of the variability for the exosome samples (panel **B**).

**Figure 4 jpm-10-00229-f004:**
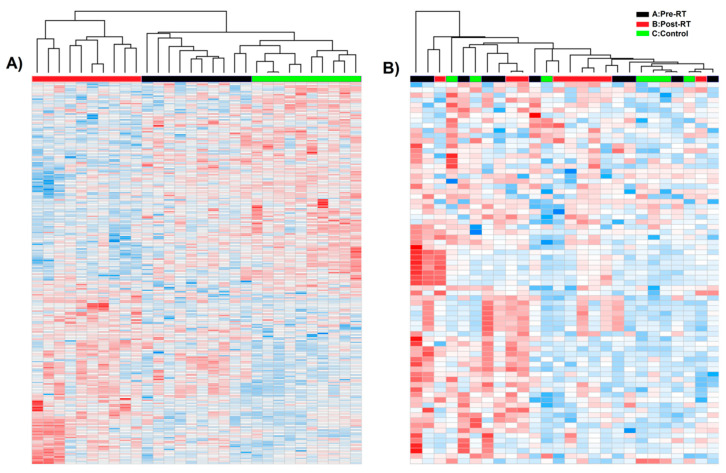
Hierarchical cluster analysis of cancer pre-RT samples A, cancer post-RT samples B, and control samples C. Shown are separate dendrograms for serum samples (panel **A**) and exosome samples (panel **B**).

**Figure 5 jpm-10-00229-f005:**
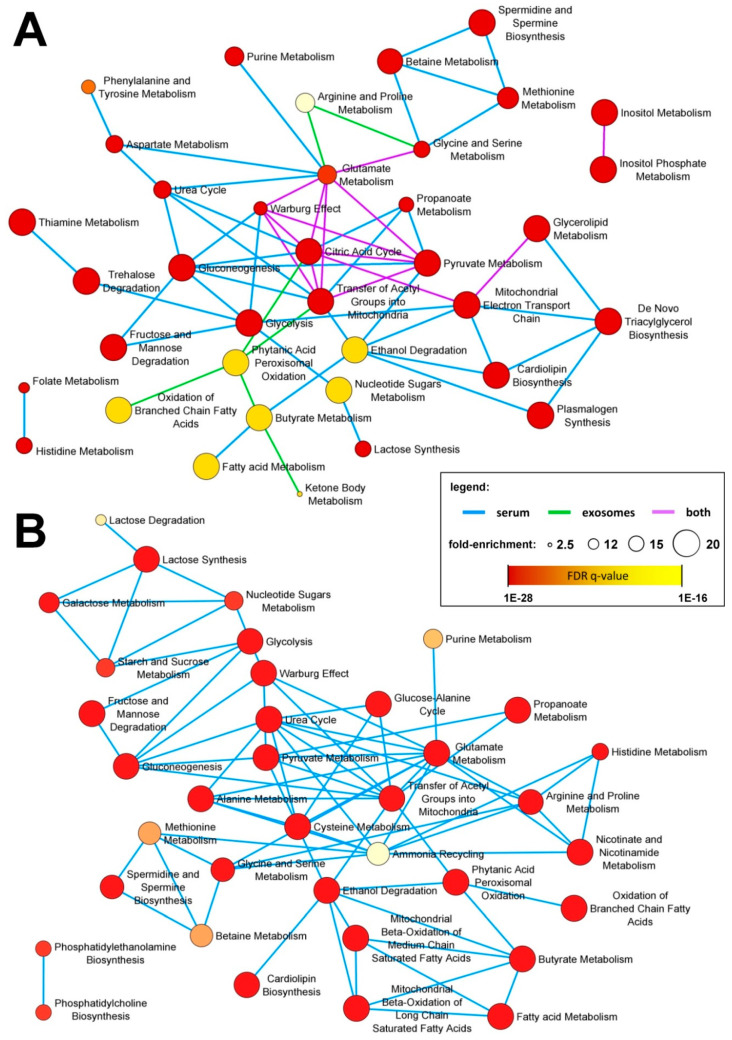
Metabolic pathways associated with specific subsets of compounds detected in the whole serum and serum-derived exosomes. Illustrated are over represented pathways associated with metabolites differentiating between cancer and control samples (panel **A**) and between pre-RT and post-RT samples (panel **B**); metabolites that showed large and medium effect size were included. The size of the network nodes corresponds to the pathway’s fold-enrichment while the statistical significance of the over-representation is color coded.

**Table 1 jpm-10-00229-t001:** Metabolites that differentiated cancer patients and healthy individuals. Listed are compounds where differences between head and neck cancer (HNC) patients (samples A) and healthy controls (samples C) showed a large effect size (RBCC effect size ≥ 0.5).

Metabolite Name	Class	Mean Abundance in Cancer (Samples A)	Mean Abundance in Control (Samples C)	Significance of Differences between Control and Cancer (RBCC Effect Size)
Serum Metabolites				
Upregulated in Cancer				
Myristic acid	Fatty acids	3.90 × 10^−3^	3.18 × 10^−3^	0.82
Hypoxanthine	Purines	3.48 × 10^−4^	1.46 × 10^−4^	0.76
L-Glutamic acid	Amino acids	4.97 × 10^−3^	2.34 × 10^−3^	0.70
Xanthine	Purines	2.96 × 10^−5^	2.01 × 10^−5^	0.66
beta-Lactose	Saccharides	2.41 × 10^−5^	9.82 × 10^−6^	0.64
L-Serine	Amino acids	8.19 × 10^−3^	6.07 × 10^−3^	0.60
Oleic acid monoglyceride	Glycerolipids	2.41 × 10^−5^	3.14 × 10^−5^	0.60
O-Acetylserine	Amino acids	4.37 × 10^−3^	3.60 × 10^−3^	0.58
Eicosenoic acid	Fatty acids	3.61 × 10^−5^	2.18 × 10^−5^	0.58
Palmitoleic acid	Fatty acids	1.02 × 10^−3^	3.12 × 10^−4^	0.56
Oleamide	Fatty acids	6.71 × 10^−5^	1.72 × 10^−5^	0.54
L-Aspartic acid	Amino acids	2.02 × 10^−3^	1.32 × 10^−3^	0.52
Downregulated in Cancer				
Inosine	Purines	4.55 × 10^−5^	4.28 × 10^−4^	−1.00
Salicylic acid	Carboxylic acids	6.74 × 10^−6^	8.44 × 10^−4^	−0.92
Adenosine	Purines	1.27 × 10^−5^	5.74 × 10^−5^	−0.89
2-Ethylhexanoic acid	Fatty acids	1.14 × 10^−4^	2.51 × 10^−4^	−0.74
Gentisic acid	Carboxylic acids	6.36 × 10^−6^	1.56 × 10^−5^	−0.64
D-Threitol	Sugar alcohols	2.01 × 10^−4^	2.88 × 10^−4^	−0.64
Oxalic acid	Carboxylic acids	2.08 × 10^−2^	2.47 × 10^−2^	−0.62
Paraxanthine	Purines	1.94 × 10^−4^	4.26 × 10^−4^	−0.62
Serotonin	Amines	5.43 × 10^−5^	1.06 × 10^−4^	−0.60
D-Ribose	Saccharides	1.50 × 10^−4^	1.51 × 10^−4^	−0.60
N-acetyl-d-hexosamine	Amines	6.21 × 10^−5^	1.69 × 10^−5^	−0.57
Nonanoic acid	Fatty acids	2.23 × 10^−4^	2.67 × 10^−4^	−0.56
D-Xylonic acid	Sugar acids	3.07 × 10^−5^	4.48 × 10^−5^	−0.56
Phosphate	Inorganic acids	1.40 × 10^−2^	1.64 × 10^−2^	−0.54
L-Isoleucine	Amino acids	2.69 × 10^−3^	3.30 × 10^−3^	−0.52
Exosome Metabolites				
Upregulated in Cancer				
1-Hexadecanol	Fatty alcohols	5.81 × 10^−5^	3.12 × 10^−5^	0.52
Downregulated in Cancer				
4-Hydroxybenzoic acid	Carboxylic acids	8.05 × 10^−7^	2.61 × 10^−5^	−0.66
Citric acid	Carboxylic acids	8.58 × 10^−6^	3.22 × 10^−4^	−0.54
Propylene glycol	Others	2.89 × 10^−5^	1.96 × 10^−4^	−0.52

**Table 2 jpm-10-00229-t002:** Metabolites that were affected by radiotherapy. Listed are compounds where differences between paired pre-RT (samples A) and post-RT (samples C) specimens showed a large effect size (Cohen’s d effect size ≥ 0.8).

Metabolite Name	Class	Mean Abundance Pre-RT (Samples A)	Mean Abundance Post-RT (Samples B)	Significance of Differences between Pre-RT and Post-RT (Cohen’s D Effect Size)
Serum Metabolites				
Upregulated by RT				
Hypotaurine	Others	6.48 × 10^−5^	1.09 × 10^−4^	−1.16
Glycerol-1-phosphate	Glycerolipids	1.03 × 10^−4^	1.46 × 10^−4^	−1.06
Oleamide	Fatty acids	6.71 × 10^−5^	1.77 × 10^−4^	−0.81
Serotonin	Amines	5.43 × 10^−5^	6.71 × 10^−5^	−0.81
Downregulated by RT				
1-Methylhistidine	Amino acids	1.13 × 10^−4^	7.84 × 10^−5^	0.96
Urea	Others	1.81 × 10^−4^	4.76 × 10^−2^	0.96
Quinic acid	Others	7.48 × 10^−5^	5.60 × 10^−5^	0.87
2-ketoglucose dimethylacetal	Hydroxy acids	1.68 × 10^−4^	7.86 × 10^−5^	0.85
4-Deoxyerythronic acid	Sugar acids	4.44 × 10^−5^	2.77 × 10^−5^	0.85
Galactosylglycerol	Glycerolipids	4.55 × 10^−5^	1.69 × 10^−5^	0.85
Gentisic acid	Carboxylic acids	6.36 × 10^−6^	3.78 × 10^−6^	0.85
D-Xylitol	Sugar alcohols	2.29 × 10^−4^	1.49 × 10^−4^	0.82

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
