# Peer review of "Metabolic Profiles of Whole Serum and Serum-Derived Exosomes Are Different in Head and Neck Cancer Patients Treated by Radiotherapy"

_jpm, 2020, doi:10.3390/jpm10040229_

Round 1

Reviewer 1 Report

This is an interesting study of metabolites identified in the serum and serum-derived exosomes of patients with head and neck cancer, prior to and following radiation treatment. The methodology of measuring metabolites in exosomes is novel. I have several comments and questions below.

  1. There are multiple grammatical errors and missing words throughout the manuscript that require correction.

As one example, there are several errors in lines 45-47: “detection in the patient’s blood of molecular fingerprint of the body response to the treatment, which could potentially enable monitoring and prediction of radiation toxicity, is another important aspect of HNC diagnostics.”

Corrections would include:

“…of a molecular fingerprint”

“..of the body’s response”

Also, the sentence is difficult to follow and better divided into two sentences.

  1. Line 65: The Warburg effect is explained in the discussion but it would be appropriate to have this explained in the introduction where it is mentioned in passing without explanation.
  2.  
  3. How is “medium effect size” being defined? Also, there is a mention in the table of how “large effect size” is defined, this should be explained in the text also.
  1. Does Figure 2 represent the results of all patients and controls? Does it include post treatment or just pre-treatment results? This should be described in the text and in the Figure legend.
  1. For the PCA analysis, what constitutes significant results? There seems to be a lot of overlapping captured even in Figure 3A in the post-RT group. In Figure 3B there seems to be a lot of overlapping of pre-RT and post-RT groups with some post-RT outliers; rather than a separation.
  1. Figure 5 should include some annotation or explanation as to the significance of each group of metabolites depicted. It is hard to interpret whether the results are meaningful otherwise.
  1. The discussion should include the reasoning behind the authors’ conclusion that there is a cancer-specific pattern metabolic profile. Details of how the results are significant should be included.
  1. What is meant in the conclusion by “the metabolite profile of serum-derived exosomes is less complex than that of the complete serum”?

Author Response

Reviewer 1

Q1. There are multiple grammatical errors and missing words throughout the manuscript that require correction. As one example, there are several errors in lines 45-47: “detection in the patient’s blood of molecular fingerprint of the body response to the treatment, which could potentially enable monitoring and prediction of radiation toxicity, is another important aspect of HNC diagnostics.” Corrections would include: “…of a molecular fingerprint”,  “..of the body’s response”. Also, the sentence is difficult to follow and better divided into two sentences.

A1. Thank you. The revised text was corrected accordingly.

Q2. Line 65: The Warburg effect is explained in the discussion but it would be appropriate to have this explained in the introduction where it is mentioned in passing without explanation.

A2. The revised manuscript was corrected accordingly. Initially, we assumed that the Warburg effect is widely described in the literature and does not require detailed description. Nevertheless, the necessary introduction was included in the revised manuscript (lanes 64-66).

Q3. How is “medium effect size” being defined? Also, there is a mention in the table of how “large effect size” is defined, this should be explained in the text also.

A3. An “effect size” is a number measuring the strength of the relationship between two variables that could be used as a complement or alternative to approaches based on the p-value for the estimation of the size of differences between groups. Different „effect size” approaches and their magnitudes (i.e., low, medium, and large) used in the current manuscript were introduced and defined in the Materials and Methods (including the relevant references). For independent samples, the rank-biserial coefficient of correlation (RBCC; an effect size equivalent of the U-Mann-Whitney test) was applied; the effect size ≥0.3 and ≥0.5 was considered medium and high, respectively [30]. For paired samples, the paired t-test derived Cohen's d effect size was applied; the effect size ≥0.5 and ≥0.8 was considered medium and high, respectively [31]. The definition of the effect size magnitude was included in the revised Results section (lanes 119, 124, 143, and 147).

Q4. Does Figure 2 represent the results of all patients and controls? Does it include post treatment or just pre-treatment results? This should be described in the text and in the Figure legend.

A4. Data illustrated in Figure 2 represent all types of samples (i.e., controls, cancer pre-RT, and cancer post-RT). This is clarified in the revised legend to Figure 2 (lanes 162-163).

Q5. For the PCA analysis, what constitutes significant results? There seems to be a lot of overlapping captured even in Figure 3A in the post-RT group. In Figure 3B there seems to be a lot of overlapping of pre-RT and post-RT groups with some post-RT outliers; rather than a separation.

A5. We obviously agree that some overlap of ovals delineating different groups of samples represented in Figure 3 exists. This overlap is particularly visible in Figure 3B, which illustrate exosome samples; this observation was noted in the text accordingly (lanes 113-115). However, this type of analysis enables just rough estimation and graphic illustration of general similarities/differences within and between groups. Therefore, the results of the PCA were followed by more specific statistical analyses. Nevertheless, the results of the unsupervised hierarchical cluster analysis of serum samples presented in Figure 4A (which corresponded to PCA data presented in Figure 3A) showed a separation of sample groups.  

Q6. Figure 5 should include some annotation or explanation as to the significance of each group of metabolites depicted. It is hard to interpret whether the results are meaningful otherwise.

A6. Figure 5 shows statistically over-represented (enriched) pathways associated with metabolites that differentiated compared groups of samples (with medium and large effect size). The size and color of nodes corresponding to specific pathways represent the fold-enrichment and its statistical significance. This is clarified in the revised legend to Figure 5.

Q7. The discussion should include the reasoning behind the authors’ conclusion that there is a cancer-specific pattern metabolic profile. Details of how the results are significant should be included.

 A7. The term “cancer-specific pattern” is related to metabolites (and associated pathways) that differentiated between control samples of healthy individuals and pre-RT samples of cancer patients. This is based on metabolites which levels showed statistically significant differences between these two groups and over-represented metabolic pathways associated with such differentiating compounds. These included metabolites associated with the Warburg effect and other pathways involved in cancer-related processes. This issue was clarified in the revised manuscript (lane 134).

A8. What is meant in the conclusion by “the metabolite profile of serum-derived exosomes is less complex than that of the complete serum”?

Q8. By this statement, we meant that a markedly lower number of compounds was detected in exosome samples than in whole serum samples. This is described in the Results section (lanes 99-102). Hence, we assumed that the larger number of different metabolites detected reflected the higher molecular complexity of the specimen. Nevertheless, we rephrased this sentence for clarity (lanes 196-198 and 318-320).

Reviewer 2 Report

This is an interesting manuscript with solid data that radiotherapy has different effects on metabolomes between serum and blood exosomes. However, the authors did not provide some insightful hypotheses for explaining the indicated data. I have some comments which might be helpful for improving the quality of the discussion section.

1. Unlike the formation of exosomal proteome, the mechanism for accumulation of metabolites in exosome is still unclear.The accumulation of metabolites in exosomes might be an uncontrolled process which results from random movement of cytoplasm metabolites into exosomes during the process of multicellular bodies or from synthesis by metabolic enzymes enclosed by the exosome. I suggest the authors to discuss this possibility.

2. In terms of sources of metabolites, serum metabolites and blood exosomal metabolites might come from different original organs. However, whether the synthesis and release of metabolites-carrying exosome is regulated by hormones are still unknown. This difference in hormone control might contribute to the difference in metabonomics between serum and exosomes. This possibility should also be discussed.

3. The changes of metabolomes can be caused by either radiotherapy itself or radiotherapy-induced change of cancer loading. The release of metabolites can be from either normal cells or residual cancer cells. The authors should mention these possibilities.

Author Response

Reviewer 2

Q1. Unlike the formation of exosomal proteome, the mechanism for accumulation of metabolites in exosome is still unclear.The accumulation of metabolites in exosomes might be an uncontrolled process which results from random movement of cytoplasm metabolites into exosomes during the process of multicellular bodies or from synthesis by metabolic enzymes enclosed by the exosome. I suggest the authors to discuss this possibility.

A1. We fully agree with this comment. However, the mechanism of accumulation of different metabolites in exomes was far beyond the scope of this work, which was focused on current exosome cargo in the context of differences between patients’ groups. Hence, assuming a short format of the manuscript, we focused on aspects directly related to the presented data. 

Q2. In terms of sources of metabolites, serum metabolites and blood exosomal metabolites might come from different original organs. However, whether the synthesis and release of metabolites-carrying exosome is regulated by hormones are still unknown. This difference in hormone control might contribute to the difference in metabonomics between serum and exosomes. This possibility should also be discussed.

A2. We fully agree with this comment. Indeed, different types of cells could produce the majority of “soluble” or “total” metabolites present in a whole serum (e.g., not only cancer cells but also hepatocytes) and extracellular vesicles circulating in the blood (e.g., not only cancer cells but also blood cells or endothelial cells). However, the concept of “liquid biopsy” is based on disease-associated molecular components that are present in biofluids irrespective of the actual source of these molecules (e.g., either cancer cells or normal cells interacting with cancer). In this proof-of-the-concept study, we compared two sets of metabolites present in the whole serum and serum-derived exosomes aimed to verify their ability to discriminate between control and cancer samples (i.e., cancer-related biomarker candidates)  as well as between pre-RT and post-RT (i.e., radiation-related biomarker candidates). We assumed that the potential discriminatory value of such metabolites is independent of their actual origin in the blood. Nevertheless, this important comment was included in the revised manuscript (lanes 209-211). 

Q3. The changes of metabolomes can be caused by either radiotherapy itself or radiotherapy-induced change of cancer loading. The release of metabolites can be from either normal cells or residual cancer cells. The authors should mention these possibilities.

A3. We agree with this comment. Several mechanisms could contribute to differences observed between pre-RT and post-RT samples. This includes response related to (i) direct effect of radiation-induced damage, (ii) activation and healing of the acute and/or late radiation toxicity, and (iii) reduced cancer load. Considering the time of post-RT samples collection (one month after the end of RT) effects related to healing of acute radiation toxicity (mechanism related to the inflammatory response in particular) and eradication of cancer cells seems the most possible. Moreover, molecules produced by either directly irradiated cells (both cancer and normal tissues) or not-irradiated cells involved in the local or systemic response to radiation could be relevant. This issue was mentioned in the revised manuscript (lanes 222-225).